# Incorporation of Amino Acid-Functionalized Ionic Liquids into Highly Porous MOF-177 to Improve the Post-Combustion CO_2_ Capture Capacity

**DOI:** 10.3390/molecules28207185

**Published:** 2023-10-20

**Authors:** Firuz A. Philip, Amr Henni

**Affiliations:** Faculty of Engineering and Applied Science, University of Regina, Regina, SK S4S 0A2, Canada; philip2f@uregina.ca

**Keywords:** CO_2_ capture, metal organic framework (MOF), ionic liquid, amino acid ionic liquid (AAIL), task-specific ionic liquid (TSIL), MOF-177

## Abstract

This study presents the encapsulation of two amino acid-based ionic liquids (AAILs), 1-ethyl-3-methylimidazolium glycine [Emim][Gly] and 1-ethyl-3-methylimidazolium alanine [Emim][Ala], in a highly porous metal–organic framework (MOF-177) to generate state-of-the-art composites for post-combustion CO_2_ capture. Thermogravimetric analysis (TGA) demonstrated a successful encapsulation of the AAILs, with a dramatic reduction in the composites’ surface areas and pore volumes. Both [Emim][Gly]@MOF-177 and [Emim][Ala]@MOF-177 had close to three times the CO_2_ uptake of MOF-177 at 20 wt.% loading, 0.2 bar, and 303 K. Additionally, 20-[Emim][Gly]@MOF-177 and 20-[Emim] [Ala]@MOF-177 enhanced their CO_2_/N_2_ selectivity from 5 (pristine MOF-177) to 13 and 11, respectively.

## 1. Introduction

Metal–organic-frameworks, often known as MOFs, are a unique class of porous materials that are currently the subject of substantial research for a broad variety of applications [1,2], notably, the storage and separation of gases, catalysis, the delivery of drugs, and energy storage [3,4,5,6,7,8]. MOFs are typically three-dimensional structures that have high surface area and porosity. These structures are formed from metal nodes that are coupled with organic linkers. MOFs have lately gained an enormous amount of popularity for applications in CO_2_ capture and storage because of their high porosity as well as their great potential for gas capture or storage due to their tunability [2,9,10,11,12]. Although some MOFs have displayed superb absorbance of CO_2_ at high pressure, in post-combustion conditions, in which the partial pressure of CO_2_ is in general about or lower than 0.15 bar, they showed very low CO_2_ uptake [13]. To enhance their CO_2_ capture capability, many strategies have been deployed including the introduction of open metal sites [14], the covalent grafting of amine functionalities to the ligands [15,16,17], the impregnation of amines [18,19,20,21], and the addition of ionic liquids (ILs) [22]. The physical impregnation of amines led to a higher CO_2_ uptake compared to the covalent grafting of amines, as a higher amine loading could be achieved by impregnation. Nonetheless, amine-impregnated sorbents are susceptible to amine loss and degradation and require a high regeneration energy [13]. Ionic liquids (ILs) have been proposed as an attractive alternative to amines partly due to their low volatility and high thermal stability; however, it was reported that in some cases, the CO_2_ uptake rather diminished for IL-modified sorbents, although improved selectivity was observed [23,24,25].

The first amine-functionalized ionic liquids (AILs) were developed by Bates et al. [26], demonstrating that the CO_2_ sorption capacity of ILs can be substantially enhanced by functionalizing the sorbent with amine groups (-NH_2_) while at the same time retaining ILs’ properties. Since then, numerous AILs, referred to as task-specific ionic liquids (TSILs), have been reported in the scientific literature. Among them, amino acid anion-functionalized ionic liquids (AAILs) have the clear advantages of an easy synthesis process, a low cost, biodegradability, and the requirement of environmentally friendly raw materials, as they are sourced from naturally occurring amino acids (AAs) [27,28,29]. As a result, AAILs are deemed promising candidates to functionalize porous MOFs and lead to advanced sorbents with high CO_2_ capture capacity. Several researchers demonstrated that the immobilization of AAILs into solid sorbents such as MCM-41, SBA-15, and PMMA led to an increase in carbon dioxide adsorption [30,31,32,33].

Recently, we showed that AAILs containing reactive amino acid (AA) anions such as glycine [Gly] and alanine [Ala] acted as effective guest molecules in a solid ZIF-8 support, significantly increasing CO_2_ uptake and CO_2_/N_2_ selectivity [34]. Hence, we aimed to expand our study and investigate new composites by impregnating these AAILs into MOF-177 as a host sorbent, which is one of the highest porous MOFs reported to date, with a pore volume of 1.69 cm^3^/g and a BET-specific surface area of over 4000 m^2^/g [35,36]. MOF-177 displays a 3D crystalline framework consisting of zinc metal clusters and 1, 3, 5-benzene tribenzoate (BTB) organic linkers. Owing to its high porosity and volume, it displayed an outstanding CO_2_ uptake performance (60.8 wt.%) at high pressure (50 bar); however, the CO_2_ uptake at low pressure (0.15 bar) was very poor (0.6 wt.%) [13]. This led us to embark on this study and incorporate AAILs into highly porous MOF-177, obtaining AAILs@MOF-177 composites with improved active sorption sites, anticipating that this strategy would lead to a higher potential for CO_2_ capture in post-combustion conditions. We are unaware of any published prior work involving these composites. In this study, we investigated the composites and compared them to pure MOF-177 in terms of thermal stability, crystal structure, and textural qualities. The produced composites’ potentials in real-world CO_2_ capture operations were investigated by examining their CO_2_ capture capacity, selectivity, and enthalpy of adsorption.

## 2. Results and Discussion

### 2.1. Characterization of the AAIL-Impregnated Sorbents

A thermogravimetric analysis of MOF-177 and the AAILs@MOF-177 composites was performed to monitor the thermal stability of the AAIL-supported MOF-177 composites from 25 to 800 °C in a nitrogen atmosphere. TGA thermograms of the composites along with those of pure AAILs and pristine MOF-177 are presented in Figure 1a–d. The thermograms of pristine [Emim][Gly] and [Emim][Ala] indicated that both AAILs were thermally stable up to 200 °C, but beyond that temperature, a sharp weight loss was observed, indicating a very rapid decomposition. According to the derivative weight loss profile shown in Figure 1b,d, it was found that the onset decomposition temperatures (T_onset_) were about 215 °C and 225 °C, respectively, whereas the pristine solid support MOF-177 was much more thermally stable than the pristine AAILs, with onset decomposition temperatures (T_onset_) of about 421 °C, in agreement with similar studies reported in the literature [36,37]. A little decrease in weight of around 2 to 4 wt.% was found for the composites at temperatures under 75 °C. This drop can most probably be attributed to the potential evaporation of any residual methanol (BP of 65 °C) that was employed during the synthesis of the composites. The thermogram of the composites revealed that the incorporated AAILs decomposed before the pristine solid support itself, as expected. However, note that there was no sharp decomposition of AAILs around their T_onset_, but rather, a gradual decomposition over the temperature range from 250 to 400 °C, which continued until MOF-177 itself started to decompose, as displayed in Figure 1b,d. This gradual decomposition was more profound at higher loadings of AAILs, as observed for 30-[Emim][Gly]@MOF-177 (Figure 1b) and 30-[Emim][Ala]@MOF-177 (Figure 1d). This progressive breakdown could be attributed to the surface association that took place between MOF-177 and the incorporated AAILs.

In an attempt to determine the effect that encapsulated AAILs had on the solid support of MOF-177, the crystal structure of the composites was investigated utilizing X-ray diffraction (XRD), and the results of this investigation are illustrated in Figure 2. In the diffractogram of pristine MOF-177, the main characteristic peaks were observed at 5.5°, 6.2°, 9.6°, 10.4°, and 11.3°. A similar pattern was reported in the literature by Li and Yang [38], and major peaks were observed at 5.2°, 4.7°, and 6.2° by Saha and Deng [37] and at 5.5° and 6.2° by Santos et al. [39]. As displayed in Figure 2a,b, the XRD of the composites revealed that the impregnation of AAILs into the MOF-177 support had a significant impact on peak intensity and position. For a low loading of 10 wt.% for both [Emim][Gly] and [Emim][Ala], the prominent characteristic peaks at 5.5°, 6.2°, and 9.6° for pristine MOF-177 showed a significant decline, but the peaks at 11.1° and 11.7° retained their intensity, with a minor shift from the original positions. It was evident that the intensity of the major peaks diminished with the increment in AAILs’ loading and eventually almost disappeared at a loading of 30 wt.%, as displayed for both 30-[Emim] [Gly]@MOF-177 and 30-[Emim][Ala]@MOF-177. This diminishment and disappearance of these peaks can be attributed to the interaction between amine groups and MOF-177-Zn. Another factor that cannot be entirely disregarded is the potential degradation of the MOF framework’s structure to some extent due to the exposure to the amine solution [40]. A similar phenomenon of a decrease in the intensity of the characteristic peaks of MOF-177 due to impregnation with polyethyleneimine (PEI) was reported by Gaikwad and co-workers [41] Their XRD data also revealed a further decline in peak intensity with the increase in the loading of PEI and eventually almost the disappearance of the peaks at 30 wt.% PEI loading. This pattern was also observed for PEI@Cu-BTC (HKUST) composites [42], polyamine@MIL-101(Cr) [43], and PEI@MCM-41 [44]. The TGA thermograms discussed in the preceding section confirmed that there was a significant interaction between AAILs and the MOF-177 support. It is important to acknowledge that additional investigation would be beneficial to examine the structural integrity of MOF-177, utilizing single-crystal XRD and scanning electron microscopy (SEM) subsequent to the exposure to a solvent under different conditions.

To further assess the impact of the incorporation of AAILs on the textural properties of the MOF-177 support, N_2_ adsorption–desorption isotherms were obtained in a liquid nitrogen environment (77 K) using the ASAP 2460 analyzer (Micromeritics) for the unmodified MOF-177 and AAILs@MOF-177 composites, and the isotherms are presented in Figure 3. The samples’ specific surface areas were determined at relative pressures (P/P_o_) between 0.04 and 0.1 bar using the Brunauer–Emmett–Teller (BET) model. In addition, the Langmuir surface area was calculated, and the results are presented in Table 1. The Hovath–Kawazoe (HK) model was used to calculate the pore size distribution for all the composites, and the results are displayed in Figure 4. The pristine MOF-177 showed a significantly higher N_2_ uptake, and the isotherm profile corresponded to a reversible type I isotherm (IUPAC classification), which was indicative of a microporous material lacking hysteresis during desorption. The BET and Langmuir surface areas of pristine MOF-177 were 4172 m^2^·g^−1^ and 4962 m^2^·g^−1^, respectively, close to the reported values by Yaghi’s group [35]. Both BET and Langmuir surface areas of MOF-177 reported in the literature varied significantly from 3100 to 4962 m^2^·g^−1^ and from 4300 to 5994 m^2^·g^−1^, respectively [35,37,38,45]. It is shown in Figure 3 that N_2_ adsorption in all composites was significantly lower than in pristine MOF-177, which is evidence of a substantial drop in surface area as well as in pore volume (Table 1). For instance, upon impregnation with 10 wt.% [Emim][Gly] and [Emim][Ala], the BET surface areas were reduced to only 187 m^2^·g^−1^ and 152 m^2^·g^−1^, respectively. The surface area and pore volume of the composites were further reduced with the addition of AAILs. The BET surface areas and pore volumes obtained for a 30 wt.% loading of [Emim][Gly] were estimated to be 27 m^2^·g^−1^ and 0.02 cm^3^·g^−1^, respectively. These results indicated that the pores were almost filled. It should be acknowledged that the [Emim][Gly]@MOF-177 composites exhibited somewhat higher surface area and pore volume compared to the [Emim][Ala]@MOF-177 composites. The reduction in surface area and pore volume confirmed that the AAILs were encapsulated into the pores of MOF-177, which was also confirmed by XRD analysis in the previous section. As the loading of AAILs rose, there was a corresponding increase in the likelihood of pore blockage, as also evident in the pore size distribution (Figure 4). Consequently, certain pores became inaccessible for N_2_ adsorption, which led to a reduction in pore volume. Similar observations were reported for PEI@UiO-66-NH_2_ [40].

### 2.2. CO_2_ Adsorption Isotherms

MOF-177 is one of the most highly porous MOFs reported to date in the open literature and, as a result, it has higher adsorption capacity at high pressure. However, its adsorption capacity in post-combustion conditions (up to 0.15 bar) is very low [13]. The current investigation involved the synthesis of composites including [Emim][Gly]@MOF-177 and [Emim][Ala]@MOF-177. These composites were prepared by introducing two amino acid anion-functionalized ionic liquids (AAILs), commonly referred to as task-specific ionic liquids (TSILs), into MOF-177 using the wet impregnation method. These composites’ equilibrium CO_2_ uptakes were determined at 30, 40, and 50 °C. The isotherms were obtained at pressures between 0.1 and 10 bar. Figure 5 shows the equilibrium CO_2_ uptake in pure MOF-177 and [Emim][Gly]@MOF-177 for pressures spanning from 0.1 to 10.0 bar (Figure 5a,c,e) and a narrower pressure range (Figure 5b,d,f) from 0.1 to 1.0 bar. The addition of [Emim][Gly] to MOF-177 increased its ability to adsorb CO_2_ at low pressures between 0.1 and 1.0 bar. When compared to those of pristine MOF-177 and all other [Emim][Gly]@MOF-177 composites at all temperatures, the CO_2_ adsorption capacity increased dramatically with the incremental addition of AAIL loading and peaked at a loading of 20 wt%. At 0.2 bar and 303 K, the CO_2_ adsorption for 20-[Emim][Gly]@MOF-177 was 0.45 mmol·g^−1^ of solid, which was three times that of pristine MOF-177 under the same circumstances. However, increasing the TSIL amount to 30 wt.% failed to result in a noticeable rise in CO_2_ uptake; rather, a decrease in CO_2_ uptake to 0.26 mmol·g^−1^ of solid was observed when compared to that observed with the 20 wt.% loading, even though it was a bit higher when compared to that of pure MOF-177 in the same conditions. It is noteworthy that the carbon dioxide absorption capacity of all [Emim][Gly]@MOF-177 composites exhibited a decrease in performance at the given temperature in comparison to pure MOF-177 when the pressure exceeded 2 bar. The CO_2_ uptake by all composites decreased when the temperature was elevated from 30 °C to 40 °C and 50 °C while maintaining the same pressure.

Figure 6 shows the equilibrium CO_2_ uptake in pure MOF-177 and [Emim][Ala]@MOF-177 for pressures spanning from 0.1 to 10.0 bar (Figure 6a,c,e) and a narrower pressure range (Figure 6b,d,f) from 0.1 to 1.0 bar. It demonstrates that the incorporation of [Emim][Ala] led to a favorable CO_2_ adsorption capability at pressures below 2 bar, similar to what was observed for the [Emim][Gly]@MOF-177 composites. The composite 20-[Emim][Ala]@MOF-177 outperformed all other [Emim][Ala]@MOF-177 composites and pure MOF-177 in CO_2_ capture capacity. This composite reached an absorption capacity of 0.42 mmol·g^−1^ of solid at 0.2 bar and 303 K, which was nearly three times higher than that of pristine MOF-177 under the same conditions. When the loading of [Emim][Ala] was raised to 30 wt.%, the CO_2_ capture capacity decreased, and the resulting values were in the middle of the capture capacity range determined for the 10 and 20 wt.% loadings. It is worth noting that under identical temperatures and pressures, the CO_2_ adsorption capacity of the [Emim][Ala]@MOF-177 composites was slightly lower than that of the [Emim][Gly]@MOF-177 composites for similar AAIL loadings.

The introduction of [Emim][Gly] and [Emim][Ala] AAILs to the MOF-177 sorbent led to a notable enhancement in CO_2_ adsorption under post-combustion circumstances (P_CO2_ ≈ 0.15 bar) due to the strong attraction between CO_2_ and the amino group present in AAILs. Previous research indicated that the amino group of AAILs reacted with CO_2_ via a process analogous to that of amines dissolving in water [26,46,47]. Wang et al. [48] suggested that as the cation and anion of [Emim][Gly] are small in size, they could come nearby an amino group and react with it to form a carbamate with a stoichiometry of 1:2 (Figure 1). Hence, it is reasonable to assume that the amino group present in these composites reacted with CO_2_ to form a carbamate, resulting in a higher CO_2_ capture capacity compared to pure MOF-177 below 1 bar.

It is noteworthy to mention that, when the pressure was below 1 bar, chemical absorption prevailed. This was attributed to the strong affinity between CO_2_ and AAILs, consequently resulting in CO_2_ absorption capacity enhancement. However, it is important to note that the occupation of the MOF-177 pores by AAILs molecules led to a significant reduction in the available surface area of the composites. Nevertheless, as the pressure increased, the composites’ advantages were lost, resulting in a CO_2_ uptake that was lower than that of pristine MOF-177 at pressures exceeding 2 bar. This decline could be due to the significant reduction in pore volume as well as the surface area of the composites. At moderate to high pressure, the absorption capacity of a sorbent is primarily influenced by physical parameters, in addition to the active chemical adsorption sites [49].

Our experimental results revealed that there was an upper limit to the loading of AAILs in relation to the enhancement of CO_2_ absorption, which was found to be 20 wt.%; beyond that limit, the CO_2_ uptake decreased. This reduction in CO_2_ capacity was ascribed to the reduction in accessible active sites in the sorbent due to the blockage created by the high AAIL concentrations, which was also evidenced by the BET surface area and pore volume results obtained for 30 wt.% AAILs@MOF-177, as discussed earlier. An analogous finding was reported by Wang et al. [48] for the impregnation of [Emim][Gly] into the nanoporous structure of polymethylmethacrylate (PMMA) by varying the loading from 0 to 100 wt.%; the optimum loading was found to be 50 wt.%. In another study, Uehara et al. [32] reported that the optimum loading of [Emim][Lys] was 60 wt.% for the mesoporous silica support SBA-15.

The stability of the composites in the CO_2_ capture operation was investigated by performing multiple cycles of adsorption at 313 K and desorption at 373 K at the atmospheric pressure in the presence of N_2_ for 20 wt.%-[Emim][Gly]@MOF-177, using an IGA microbalance. The obtained results are displayed in Figure 7. They showed that the composite sorbent could almost maintain its original adsorption capacity during multicycle operations. In addition, the CO_2_ uptake process for AAILs@MOF-177 was completely reversible, suggesting that the composites could be readily regenerated in the presence of flowing N_2_ at 100 °C.

### 2.3. Selectivity for CO_2_/N_2_

To be deemed effective in the post-combustion capture of carbon dioxide (CO_2_), a solid sorbent must have a notable level of selectivity towards CO_2_ compared to nitrogen (N_2_). Consequently, the determination of N_2_ adsorption isotherms at 40 °C was undertaken to quantify the CO_2_/N_2_ selectivity. Each isotherm encompassed a range of pressures ranging from 0.1 to 10 bar. In our current study, we employed a particular methodology to determine the optimal selectivity. Our strategy entailed calculating the selectivity by comparing the molar uptakes of individual components at a given pressure, according to Equation (1) [2]:(1)$$S=\frac{q_{CO2}}{q_{N2}}$$
where S denotes the selectivity, and q_CO2_ and q_N2_ represent the molar uptakes of CO_2_ and N_2_, respectively. Figure 8 presents the results of the calculations of CO_2_/N_2_ selectivity for both [Emim][Gly]@MOF-177 and [Emim][Ala]@MOF-177 composites. Throughout the whole pressure spectrum, the CO_2_/N_2_ selectivity of pure MOF-177 fluctuated between the values of 3 and 5. It was discovered that the impregnation of [Emim][Gly] and [Emim][Ala] into MOF-177 boosted the selectivity for all loadings compared to that of the virgin MOF-177 up to a pressure of 2 bar. As for [Emim][Gly]@MOF-177, it was observed that the compound 20-[Emim][Gly]@MOF-177 demonstrated the highest selectivity, reaching a value of about 13 at a pressure of 0.2 bar and a temperature of 313 K. Nevertheless, the observed value exhibited a gradual decline as the pressure was raised, as depicted in Figure 8a. It is important to note that the increase in AAIL loading to 30-[Emim][Gly] did not increase the selectivity; rather, it resulted in a selectivity that was worse than that observed for the 10 wt.% loading, except at 0.1 bar. A similar behavior was seen for the [Emim][Ala]@MOF-177 composites, with maximum selectivity of around 15 (0.1 bar) and 11 (0.2 bar) displayed by 20-[Emim][Ala]@MOF-177. It was discovered, however, that the selectivity of the 30-[Emim][Ala]@MOF-177 composite was nearly identical to that of the 20-[Emim][Ala]@MOF-177 composite.

The encapsulated amino acid-base ionic liquids contributed to the enhancement of CO_2_/N_2_ selectivity. This improvement was observed at pressures below 2 bar. As previously discussed, it was proposed that the presence of amino acids leads to the formation of active chemical sorption sites for CO_2_, facilitating the creation of an N-C bond. This interaction is similar to that observed when CO_2_ interacts with an aqueous amine solution [46,48]. In our work, it resulted in additional CO_2_ capture, although the surface areas and pore volumes were reduced due to the addition of the ionic liquid. On the other hand, N_2_ did not have an affinity for the amino group as the adsorption was physical and depended on the available surface area. Hence, CO_2_ uptake was dominant at low pressure compared to N_2_ uptake, resulting in higher CO_2_/N_2_ selectivity. However, as the pressure increased, the physical adsorption sites also became determining factors of the adsorption capacity besides the active chemical adsorption sites in the sorbent. Consequently, the CO_2_/N_2_ selectivity of the composites decreased as the pressure increased and became lower than that of pristine MOF-177 at pressures above 2 bar.

The CO_2_/N_2_ selectivity and CO_2_ uptake of the examined composites, along with those reported in our previous studies and for other composites in the literature, are presented in Table 2. It is evident that the CO_2_ uptake and CO_2_/N_2_ selectivity of the best-performing composite, AAILs@MOF-177 (20%), were lower than those of our previously reported best-performing AAIL composites containing a ZIF-8 support (30%). These composites also performed better than some of the reported composites in the literature. An even better performance was expected for AAILs@MOF-177, given the porosity of pristine MOF-177; however, the host–guest interactions in a composite, the guest molecules’ loading, the support’s stability, the composite’s porosity significantly influence the CO_2_ adsorption capacity.

### 2.4. Equilibrium Isotherm Modeling

To accurately represent the outcomes of the experimental investigations pertaining to the design of adsorption and desorption processes, it was imperative to construct a model of the equilibrium isotherm. The presence of encapsulated AAILs within the pore structure of MOF-177 resulted in the composite material under investigation exhibiting binding sites of varying strengths. After conducting a comparative analysis with various existing models, it was determined that the dual-site Langmuir (DSL) model [24,50] had a high level of suitability. The model incorporates the Langmuir adsorption mechanism, which involves two distinct adsorption sites. The overall adsorption is determined by the cumulative adsorption, involving the absorption occurring at each individual site, as represented by Equation (2) [1]:(2)$$N_{e}=\frac{N_{A}b_{A}P}{1+b_{A}P}+\frac{N_{B}b_{B}P}{1+b_{B}P}$$

In the DSL model, the equilibrium intake of CO_2_ (*N_e_*) is denoted in millimoles per gram of solid, while the equilibrium pressure (*P*) is given in bar. The parameters *N_A_*, *N_B_*, *b_A_*, and *b_B_* were regressed using the model. Their values are presented in Table 3 and Table 4. These tables include the parameters obtained from the regression analysis for different loadings of ionic liquids (ILs) at various temperatures. Figure 9 and Figure 10 provide visual representations of the fitting curves generated by the DSL model. Due to the R^2^ values exhibiting a high degree of proximity to unity, the model effectively achieved a remarkable level of conformity with the experimental data. Consequently, the data obtained from the model were utilized to compute the heat of adsorption in the subsequent section.

### 2.5. Isosteric Heat of Adsorption (Q_st_)

The determination of the adsorption enthalpy of carbon dioxide (*Q_st_*), referred to as the isosteric heat of adsorption, plays a pivotal role in the adsorption process. It displays the gas molecules’ affinity for the adsorbents and the degree of their interaction. The energy requirements for the adsorption–desorption process were therefore quantified. The CO_2_ isotherms at (303, 313, and 323) K were used to calculate the adsorption enthalpy (*Q_st_*). At first, the DSL model was employed to establish a suitable match for the isotherms, as discussed in the preceding section. Subsequently, the Clausius–Clapeyron Equation (3) was utilized [1]
(3)ln⁡PN=−QstR1T+C

In the given context, the symbol *P* denotes the pressure in bar, *N* represents the extent of CO_2_ adsorption, *T* signifies the temperature measured in Kelvin (K), and *R* denotes the universal gas constant. The equation was utilized to produce graphs depicting the natural logarithm of the partial pressure (ln *P*) as a function of the reciprocal of temperature (1/*T*) while maintaining a constant rate of carbon dioxide consumption. The value of *Q_st_* was then determined by calculating the slope of these plots. The findings are depicted in Figure 11, illustrating the outcomes for both pure MOF-177 and AAILs@MOF-177.

The *Q_st_* values for the pure MOF-177 remained relatively stable at 13 kJ·mol^−1^. In contrast, a significant rise in *Q_st_* was observed for the composites, particularly at low levels of CO_2_ uptake. When compared to other [Emim][Gly] composites, 20-[Emim][Gly]@MOF-177 showed the highest values for *Q_st_*, reaching a maximum of −28 kJ·mol^−1^ at 0.2 mmol·g^−1^ CO_2_ uptake. This value was double that of pure MOF-177 under identical conditions (Figure 11a). Similarly, it was observed that the composite material 20-[Emim][Ala]@MOF-177 displayed the highest *Q_st_* values compared to the other [Emim][Ala]@MOF-177 composites (Figure 11b). The significant increase in *Q_st_* can be ascribed to the strong intermolecular forces between carbon dioxide (CO_2_) and the ionic liquids that had been incorporated within the pores of MOF-177. There exists a hypothesis suggesting that the anions of AAILs, containing the -NH_2_ group, undergo a reaction to form an N-C bond. This reaction is believed to contribute to the higher heat release observed after the adsorption of CO_2_ [48]. The decrease in the quantity of available adsorption sites can be attributed to the observed decline in the *Q_st_* value across all composites, which coincided with an increase in CO_2_ uptake. A similar observation was reported for a composite of MIL-100 (Fe) modified with DETA [18]. As expected, the *Q_st_* values for both 30%-[Emim][Gly]@MOF-177 and 30%-[Emim][Ala]@MOF-177 were between those for the 10 and 20 wt.% loadings, which confirmed the CO_2_ adsorption isotherm and selectivity pattern observed and discussed in the previous section.

## 3. Materials and Methods

### 3.1. Materials

Sigma Aldrich supplied methanol (CAS 67-56-1), [Emim][Gly] (CAS 766537-74-0), [Emim][Ala] (CAS 766537-81-9), and MOF-177 (basolite Z377, CAS: 676593-65-0). Before beginning the sample preparation, MOF-177 was allowed to dry overnight at 110 °C. All other compounds were used as supplied. To reduce the amount of time that the samples were exposed to moisture, the ionic liquid, MOF-177, and the produced composites were stored in a glovebox (Clean Tech LLC, Minot, ND, USA) filled with argon gas. Praxair Inc. Canada (Mississauga, ON, Canada) was the supplier of CO_2_ and N_2_, which had a high purity level (99.99 vol.%).

### 3.2. Preparation of the AAIL@MOF-177 Composites

Using the wet impregnation approach and methanol as a solvent, amino acid ionic liquids [Emim][Gly] and [Emim][Ala] were effectively embedded within the porous MOF-177 support. In summary, the desired amount of AAILs was added to 4 mL of methanol in a small glass vial, and the mixture was homogenized by shaking it for 30 min. The AAIL–methanol solution was carefully added in a dropwise manner to the pre-weighed dehydrated MOF-177, which was contained in a separate glass vial. The resulting combination was then agitated for 1 h. Following 24 h of solvent evaporation under ambient conditions, any residual solvent was removed by subjecting the composite to a drying process at a temperature of 80 °C, well above the boiling point of methanol (65 °C), for 2 h. The created composite samples were stored in an argon-filled glovebox to prevent exposure to moisture and were labelled as X-AAILs@MOF-177, in which X represents the weight % of AAILs used; for instance, 10 wt.% [Emim][Gly] is here referred to as 10-[Emim][Gly]@MOF-177.

### 3.3. Characterization

The thermogravimetric analysis (TGA) of [Emim][Gly], [Emim][Ala], pristine MOF-177, and all synthesized AAILs@MOF-177 composites was conducted with a Shimadzu Thermal Gravimetric Analysis device (TGA-50). The analysis was performed using a nitrogen flow rate of 50 mL/min with the temperature gradually increasing to 800 °C at a rate of 10 °C/min. In each of the analyses, a sample amount of roughly 10 to 12 mg was utilized. To analyze the crystal structure of the pure MOF-177 support as well as of the AAILs@MOF-177 composites, a tabletop X-ray diffraction (XRD) device (Rigaku Miniflex-II) was employed. The Cu Ka radiation used in this experiment had a wavelength of 1.5418 Å. The examination was carried out at a scanning step of 1.2 °C/min between 2θ values of 2 and 20 degrees, a temperature of 77 K, and using an instrument manufactured by Micromeritics ASAP. The N_2_ adsorption and desorption isotherms of MOF-177 and the as-synthesized composites were determined. Based on the data from the N_2_ isotherm, the textural features of each sample, such as the surface area (BET and Langmuir) as well as the pore volume, were computed.

### 3.4. Adsorption Isotherms

An intelligent gravimetric analyzer (IGA-003), manufactured by Hidden Isochema Ltd. (Warrington, UK), was utilized to acquire CO_2_ adsorption data at 303, 313, and 323 K, as well as N_2_ adsorption data at 313 K, throughout a wide span of pressures ranging from 0.1 bar to 10 bar. The IGA consisted of a computer-controlled microbalance that measured the weight in real time with a precision of 1 µg. Between 50 and 70 mg of material were deposited into a sample bucket for each isotherm. After heating the sample chamber to 80 °C using a water bath and vacuuming to 10 mbar with a diaphragm and turbo-pump (Pfeiffer, Asslar, Germany), the sample weight stayed constant for 1 h, indicating that all the solvent, moisture, and contaminants were removed. Following the completion of the outgassing process, the temperature of the water bath was adjusted to the predetermined isotherm temperature, and subsequently, the sample was permitted to attain the designated temperature. Once the sample was ready, the pressure level was pre-set to a value ranging from 0.1 to 10 bar in the IGASwin software (v.1.03.143), and the isotherm measurements were initiated. A mass flow controller (MFC) was used to regulate the quantity of CO_2_ or N_2_ injected into the chamber to maintain the desired pressure. The IGASwin program kept track of the real-time measurements of mass, temperature, and pressure. After allowing each pressure level to establish equilibrium for at least two hours and recording the results, more CO_2_ or N_2_ was injected via the MFC at the following pressure level. At a given temperature, this procedure was repeated for each of the predetermined pressures. After the experiment was completed, the buoyancy effect was accounted for in the real-time adsorption data.

### 3.5. Cyclic Adsorption–Desorption Test

At 40 °C and 1 bar, the IGA was used to perform cyclic CO_2_ adsorption and desorption studies. As explained earlier, first, the adsorption sample was degassed, and the temperature was adjusted to 40 °C. After 30 min to achieve temperature stabilization, the injection of carbon dioxide (100 mL/min) was carried out to commence the process of adsorption, maintained for 60 min. To facilitate the desorption, the sample was thereafter exposed to a thermal treatment at a temperature of 100 °C under a continuous flow of nitrogen gas (100 mL/min) for 150 min. This process aimed at eliminating the adsorbed carbon dioxide from the sample. These processes of adsorption and desorption were carried out for five cycles.

## 4. Conclusions

Enhanced CO_2_ adsorption and selectivity were observed in composites made by encapsulating two amino acid-based ionic liquids (AAILs) within the highly porous metal-organic framework MOF-177. The composite material with a loading of 20 wt.% [Emim][Gly] demonstrated the highest recorded CO_2_ uptake of 0.45 mmol·g^−1^ of solid at a pressure of 0.2 bar and a temperature of 303 K. The composite material with 20 wt.% [Emim][Ala] exhibited a CO_2_ uptake of 0.42 mmol·g^−1^ of solid, three and 2.8 times higher than the CO_2_ uptake of pure MOF-177. The introduction of AAILs resulted in an enhancement of the CO_2_/N_2_ selectivity, with values increasing from 5 (for pure MOF-177) to 13 for [Emim][Gly] and 11 for [Emim][Ala] at a pressure of 0.2 bar and a temperature of 313 K. The interaction between carbon dioxide and the amino (-NH_2_) functional group, facilitated by the anion of the amino acid ionic liquids (AAILs), resulted in increased adsorption enthalpy (*Q_st_*) values. The *Q_st_* values for pure MOF-177 remained relatively stable at 13 kJ·mol^−1^; in contrast, a significant rise in *Q_st_* was observed for the composites, particularly at low levels of CO_2_ uptake. When compared to other [Emim][Gly] composites, 20-[Emim][Gly]@MOF-177 showed the highest *Q_st_* values, reaching a maximum of −28 kJ·mol^−1^ at 0.2 mmol·g^−1^ of CO_2_ uptake. The present investigation also revealed that the ideal loading of AAIL was 20 wt.%, whereas any subsequent increase in loading to 30 wt.% was inadvisable. At a loading of 30 wt.%, the decline in CO_2_ absorption could potentially be attributed to a reduction in the availability of active sites on the sorbent. This reduction could be caused by the blockage or a partial collapse of the MOF-177 structure, as evidenced by a decrease in both surface area and pore volume.

This study provides insights into the structural integrity of AAILs@MOF-177 composites, their performance in terms of CO_2_ capture and CO_2_/N_2_ selectivity, and their adsorption enthalpies in post-combustion CO_2_ capture processes.

## Data Availability

The data presented in this study are available on request from the corresponding author.

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
