# Peer review of "Incorporation of Amino Acid-Functionalized Ionic Liquids into Highly Porous MOF-177 to Improve the Post-Combustion CO_2_ Capture Capacity"

_molecules, 2023, doi:10.3390/molecules28207185_

Round 1

Reviewer 1 Report

REVIEWER REPORT

This article entitled “Incorporation of Functionalized Amino Acid Ionic Liquid into Highly Porous MOF-177 to Improve Post-Combustion CO2 Capture Capacity” written by Firuz A. Philip and Amr Henni describe an efficient procedure to encapsulate two amino acid-based ionic liquids (AAILs), 1-Ethyl-3-methylimidazolium glycine [Emim][Gly] and 1-Ethyl-3-methylimidazolium alanine [Emim][Ala], in the metal organic framework (MOF-177) based on zinc metal clusters and 1,3,5-benzene tribenzoate (BTB) organic linkers. The resulting materials [Emim] [Gly]@MOF-177 and [Emim][Ala]@MOF-177 were interestingly applied for post-combustion CO2 capture at 20 wt.% loading, 0.2 bar, and 303 K, showing significant improvement in CO2 capture and CO2/N2 selectivity compared to the pristine MOF-177. This manuscript is well organized and supported. Therefore, I recommend the publication of this manuscript in Molecules journal after minor revisions:

1-      Include references after this sentence in the introduction section: “notably the storage and separation of gases, catalysis, and the delivery of drugs.” The referee suggests the following:

[10.1016/j.mattod.2017.07.006]

[10.1016/j.ensm.2015.11.005]

[10.1016/j.ultsonch.2018.08.027]

[10.1021/acs.chemrev.7b00091]

[10.1016/j.micromeso.2023.112565].

Additionally, introduce energy as a relevant application of MOFs.

2-      Include some reference to justify this affirmation: “Although some MOFs have displayed superb adsorbance of CO2 at high pressure, at post-combustion conditions, in which the partial pressure of CO2 is in general only about or lower than 0.15 bar, they showed very low CO2 uptake”.

3-      Referring to the work previously reported by the authors [10.1016/j.micromeso.2021.111580] where they use the MOF ZIF-8 carrying out an analogous process of AAILs, why in this case the crystallinity of the MOF-177 undergoes such a marked change with the increased load of AAILs? "The existence of the guest molecules and their interaction with the host solid support account for the observed behavior" should also occur in the case of ZIF-8. More evidence should be indicated to clarify these differences. Furthermore, the authors incorrectly indicate that morphological changes occur but what really happens are changes in the crystalline structure of the MOF.

4-      In line with the previous point, the reviewer requested the incorporation of scanning electron microscopy (SEM) images into the manuscript to analyze the morphological changes that occur in the material when the AAILs were embedded within the MOF-177 support.

5-      The surface area of MOF-177 after the addition of AAILs decreases impressively. Reviewing previous work with the ZIF-8 MOF [10.1016/j.micromeso.2021.111580], this behavior does not occur when AAIL loads are increased. Why do these differences exist? The authors explain that this can be attributed to possible structural and morphological changes due to the collapse of some of the MOF-177 structure. Therefore, can the composite continue to be called MOF in the rest of the manuscript? More details should be provided by the authors to clarify this matter.

6-      Are there changes in the average pore size? The reviewer suggests incorporating pore size distribution graphs for each of the materials.

7-      The authors correctly explain the mechanism of CO2 adsorption under post-combustion circumstances. However, laying out a figure in schematic form would greatly help to visualize how this process is carried out.

Minor editing of English language required

Author Response

Comments and Suggestions for Authors

We thank the 3 reviewers so much for reviewing our manuscript for publication and for the suggestion for some revisions. We have revised the entire manuscript according to all the reviewers' suggestions. In this revised version, the introduction, characterization, results and discussion were all revised and further elaborated. Additional references were added to support our conclusions. Graph 4 for pore size distribution was added, scheme 1 for the proposed reaction and Table 2 for the comparison with similar composites reported in previous literature were also added. All the changes are highlighted in green. We have addressed your suggestions to the best of our knowledge.

REVIEWERs REPORT\

Reviewer # 1

This article entitled “Incorporation of Functionalized Amino Acid Ionic Liquid into Highly Porous MOF-177 to Improve Post-Combustion CO2 Capture Capacity” written by Firuz A. Philip and Amr Henni describe an efficient procedure to encapsulate two amino acid-based ionic liquids (AAILs), 1-Ethyl-3-methylimidazolium glycine [Emim][Gly] and 1-Ethyl-3-methylimidazolium alanine [Emim][Ala], in the metal organic framework (MOF-177) based on zinc metal clusters and 1,3,5-benzene tribenzoate (BTB) organic linkers. The resulting materials [Emim] [Gly]@MOF-177 and [Emim][Ala]@MOF-177 were interestingly applied for post-combustion CO2 capture at 20 wt.% loading, 0.2 bar, and 303 K, showing significant improvement in CO2 capture and CO2/N2 selectivity compared to the pristine MOF-177. This manuscript is well organized and supported. Therefore, I recommend the publication of this manuscript in Molecules journal after minor revisions:

1-      Include references after this sentence in the introduction section: “notably the storage and separation of gases, catalysis, and the delivery of drugs.” The referee suggests the following:

[10.1016/j.mattod.2017.07.006]

[10.1016/j.ensm.2015.11.005]

[10.1016/j.ultsonch.2018.08.027]

[10.1021/acs.chemrev.7b00091]

[10.1016/j.micromeso.2023.112565]. 

Additionally, introduce energy as a relevant application of MOFs.

Ans: References included, and manuscript was revised accordingly.

2-      Include some reference to justify this affirmation: “Although some MOFs have displayed superb adsorbance of CO2 at high pressure, at post-combustion conditions, in which the partial pressure of CO2 is in general only about or lower than 0.15 bar, they showed very low CO2 uptake”.

Ans: Reference is included, and the manuscript was revised accordingly. This reference has compiled information for various MOF performances.

3-      Referring to the work previously reported by the authors [10.1016/j.micromeso.2021.111580] where they use the MOF ZIF-8 carrying out an analogous process of AAILs, why in this case the crystallinity of the MOF-177 undergoes such a marked change with the increased load of AAILs? "The existence of the guest molecules and their interaction with the host solid support account for the observed behavior" should also occur in the case of ZIF-8. More evidence should be indicated to clarify these differences. Furthermore, the authors incorrectly indicate that morphological changes occur but what really happens are changes in the crystalline structure of the MOF.

Ans: Authors agree with the reviewer that the observed XRD pattern is due to the interaction and not the morphological change. Almost identical observations were reported in multiple publications which is evident in the following journals supplementary. We have updated the manuscripts and added all the references. ZIF8 is know as one of the most stable MOF compared to the MOF-177 which might explain the difference.

doi.org/10.1016/j.jece.2021.105523

DOI: 10.1039/c4ta01174k

4-      In line with the previous point, the reviewer requested the incorporation of scanning electron microscopy (SEM) images into the manuscript to analyze the morphological changes that occurred in the material when the AAILs were embedded within the MOF-177 support.

Ans: The authors also agree with the suggestions. Actually, we also felt during the characterization analysis that SEM would be a great addition. However, our university did not have the facility and hence approached some other institutes but could not get it done due to unavailability of instrument post COVID that time.

As discussed in point 3, we also do not expect much morphological change as eluded in aforementioned journals. However, we aim to add SEM analysis in our future study. We have added similar suggestions in the manuscript. Note that in all our previous seven studies, we did not use SEM but rather FTIR and XRD.

doi.org/10.1016/j.jece.2021.105523

5-      The surface area of MOF-177 after the addition of AAILs decreases impressively. Reviewing previous work with the ZIF-8 MOF [10.1016/j.micromeso.2021.111580], this behavior does not occur when AAIL loads are increased. Why do these differences exist? The authors explain that this can be attributed to possible structural and morphological changes due to the collapse of some of the MOF-177 structure. Therefore, can the composite continue to be called MOF in the rest of the manuscript? More details should be provided by the authors to clarify this matter.

Ans: Although the pore volume and surface area for zif8 decreased with the increase of loading for MOF-177 it was greater. The changes are more applicable to pore filling and blockage of the pore due to the occupation of AAILs which is evident with a closer look at the pore size distribution. In the case of ZIF some pore widening effect was observed which was not the case for MOF177. The manuscript has be revised.

6-      Are there changes in the average pore size? The reviewer suggests incorporating pore-size distribution graphs for each of the materials.

      Ans: As discussed in the earlier point, pore size distribution has been added to the manuscripts.

7-      The authors correctly explain the mechanism of CO2 adsorption under post-combustion circumstances. However, laying out a figure in schematic form would greatly help to visualize how this process is carried out.

      Ans: A schematic of reactions is added as scheme 1 in the manuscript and references are added for the proposed mechanism.

Reviewer 2 Report

This present manuscript reported loading two ionic liquids (AAILs) into MOF-177. The different loading amounts of AAILs were studied. Finally, the obtained materials were applied for CO2 and selective CO2/N2 at different adsorption conditions. The excellent capacity and selectivity at low pressure was an advantage over the pristine MOF-177 (no ILs loading). The author may consider the comment below before getting published in Molecules.

1. XRD pattern: The pristine pattern of MOF-177 indicated that the crystal disappeared after first loading AAILs (10%). The spectra of MOF-177 were degraded entirely at the first AAILs loading, which could imply structural decomposition after loading ILs. On the other hand, new peaks were observed in the diffraction of synthesized materials. Therefore, I'm afraid I have to disagree with the authors mentioning that "…but peaks at 11.1 and 11.7 retained their intensity with a minor shift from the original positions." on lines 111-113.

2. Besides evaluating the structural change of materials, morphology characterization via SEM could be performed on synthesized materials.

3. The coordination or bonding between AAILs and MOF-177 needs discussion. The characterization of functional groups on synthesized materials should be performed or purposed the mechanism of coordination.

4. At present, AAILs are essential for improving the CO2 adsorption capacity and selectivity(at low-pressure adsorption). The host support (MOF-177) lost the function of porosity (the characterization was also confirmed in the loss of crystal structure and porosity after loaded ILs) of materials. In addition, the authors also mentioned the role of ILs over MOF-177 in lines 222-231. Therefore, what is the function of MOF-177 to play a role in the key idea of this work? Using more stable support (such as meso-silica or zeolites) may be better (more stable, low cost, etc.) than using MOF-177.  

5. Authors have published relatively reported works in recent years. Authors could illustrate the differences between the present work and those early reported articles, such as the difference, advantage, improvement, application, performance, or stability.  

- Mohamedali, Mohanned, Amr Henni, and Hussameldin Ibrahim. "Investigation of CO 2 capture using acetate-based ionic liquids incorporated into exceptionally porous metal–organic frameworks." Adsorption 25 (2019): 675-692.

- Philip, Firuz A., and Amr Henni. "Incorporation of Functionalized Amino Acid Ionic Liquid into Highly Porous MOF-177 to Improve Post-Combustion CO2 Capture Capacity." (2023).

- Mohamedali, M.; Henni, A.; Ibrahim, H. Investigation of CO2 Capture Using Acetate-Based Ionic Liquids Incorporated into 556

Minor revisions from native English may suggest improving the manuscript. 

Author Response

Reviewer # 2

Comments and Suggestions for Authors

This present manuscript reported loading two ionic liquids (AAILs) into MOF-177. The different loading amounts of AAILs were studied. Finally, the obtained materials were applied for CO2 and selective CO2/N2 at different adsorption conditions. The excellent capacity and selectivity at low pressure was an advantage over the pristine MOF-177 (no ILs loading). The author may consider the comment below before getting published in Molecules.

  1. XRD pattern: The pristine pattern of MOF-177 indicated that the crystal disappeared after first loading AAILs (10%). The spectra of MOF-177 were degraded entirely at the first AAILs loading, which could imply structural decomposition after loading ILs. On the other hand, new peaks were observed in the diffraction of synthesized materials. Therefore, I'm afraid I have to disagree with the authors mentioning that "…but peaks at 11.1 and 11.7 retained their intensity with a minor shift from the original positions." on lines 111-113.

Ans: Almost identical observations were reported in multiple publications which is evident in the following journals supplementary. We have updated the manuscripts and added all the references. However, the authors also acknowledge that there is the possibility of degradation to some extent as referred to in other similar work which has been also added in the manuscripts.

doi.org/10.1016/j.jece.2021.105523

DOI: 10.1039/c4ta01174k

  1. 2. Besides evaluating the structural change of materials, morphology characterization via SEM could be performed on synthesized materials.

Ans: The authors agree with the suggestion. Actually, we also felt during the characterization analysis that SEM would be a great addition. However, our university do not have the facility and hence approached some other institute but could not get it done due to the unavailability of instrument post-COVID time.

As discussed, with the first reviewer, we also do not expect much morphological change as eluded in the aforementioned journal. However, we aim to add SEM analysis in our future studies. We have relied in our multiple published studies on XRD, TGA and FTIR. We have added similar suggestions in the manuscript.

doi.org/10.1016/j.jece.2021.105523

  1. The coordination or bonding between AAILs and MOF-177 needs discussion. The characterization of functional groups on synthesized materials should be performed or purposed the mechanism of coordination.

Ans: Authors agree that addition of a computational study (MD) to elucidate the interaction between AAILs and MOF-177 would be a great addition as the following paper.

doi.org/10.1016/j.ces.2015.10.003

However, unfortunately, this is beyond the scope of this study, at the moment. We could not find literature which dealt with AAILs and MOF-177 composite. It would be a great future work to shed more light on it.  We have planned to add a new PhD student to join the research group in Winter 2024 and will only work on the computational study of Functionalized MOFs.  We have done similar work dealing with the kinetics of CO2 absorption with amines and the prediction of pKa with amines but not within solid material.

  1. At present, AAILs are essential for improving the CO2adsorption capacity and selectivity (at low-pressure adsorption). The host support (MOF-177) lost the function of porosity (the characterization was also confirmed in the loss of crystal structure and porosity after loaded ILs) of materials. In addition, the authors also mentioned the role of ILs over MOF-177 in lines 222-231. Therefore, what is the function of MOF-177 to play a role in the key idea of this work? Using more stable support (such as meso-silica or zeolites) may be better (more stable, low cost, etc.) than using MOF-177.

Ans. We expected better performance from AAILs@MOF-177 given the porosity of the pristine MOF177. Although composites performed better than the pristine but compared to our previous works with ZIF-8, it is less. It is also noted that these composites performed better than some of the reported composites in the literature (comparison added in the manuscripts). The authors agree that now that the research is performed, this might not be the best MOF in the market as AAILs@ZIF8 performed better, however this works gives insight of the performance of MOF-177 support. We have updated the narrative to reflect that.

  1. Authors have published relatively reported works in recent years. Authors could illustrate the differences between the present work and those early reported articles, such as the difference, advantage, improvement, application, performance, or stability.  

- Mohamedali, Mohanned, Amr Henni, and Hussameldin Ibrahim. "Investigation of CO 2 capture using acetate-based ionic liquids incorporated into exceptionally porous metal–organic frameworks." Adsorption 25 (2019): 675-692.

- Philip, Firuz A., and Amr Henni. "Incorporation of Functionalized Amino Acid Ionic Liquid into Highly Porous MOF-177 to Improve Post-Combustion CO2 Capture Capacity." (2023).

- Mohamedali, M.; Henni, A.; Ibrahim, H. Investigation of CO2 Capture Using Acetate-Based Ionic Liquids Incorporated into 556

 Ans: We added a comparison table in the manuscript including the mentioned journals  in addition to other composites.

Reviewer 3 Report

Philip and coworkers present an interesting work by encapsulation of amino acid-based ionic liquids into the highly porous MOF-177 to realize better adsorption of CO2 and high CO2/N2 selectivity. The authors synthesized and fully characterized the post-synthesis modified MOF-177 and conducted complete gas adsorption and selectivity experiments; the manuscript is also well written. In sum, after addressing the following minor issues, I recommend this work to be published. 

1) For TGA curves, if the authors confirm the weight loss <100C is from the methanol, the authors should try to remove the solvent by activating the MOF to collect a clean TGA curve to support their conclusion of better stability. 

2) It seems the PXRD results cannot support the hypothesis of the modified MOFs retain the same/similar crystallinity as the pristine MOF-177 due the loss of low-angle peaks. The authors should better explain the PXRD or conduct morphology study, like SEM, to support the retain of the crystallinity.

Author Response

Reviewer # 3

Philip and coworkers present an interesting work by encapsulation of amino acid-based ionic liquids into the highly porous MOF-177 to realize better adsorption of CO2 and high CO2/N2 selectivity. The authors synthesized and fully characterized the post-synthesis modified MOF-177 and conducted complete gas adsorption and selectivity experiments; the manuscript is also well written. In sum, after addressing the following minor issues, I recommend this work to be published. 

  • For TGA curves, if the authors confirm the weight loss <100C is from the methanol, the authors should try to remove the solvent by activating the MOF to collect a clean TGA curve to support their conclusion of better stability.

Ans: We did activation at 80 C. We agree, it would be better for future works to perform the activation about 100C to make sure remaining or any trace solvent is gone.

  • It seems the PXRD results cannot support the hypothesis of the modified MOFs retain the same/similar crystallinity as the pristine MOF-177 due the loss of low-angle peaks. The authors should better explain the PXRD or conduct morphology study, like SEM, to support the retain of the crystallinity.

Ans : Almost identical observation were reported in multiple publication which is evident in the following journals supplementary. We have updated the manuscripts and added all the references. However, the authors also acknowledge that there is possibility of degradation to some extent as referred in other similar work which has been also added in the manuscript.

doi.org/10.1016/j.jece.2021.105523

DOI: 10.1039/c4ta01174k

As discussed in with reviewer 1, we also do not expect much morphological change as eluded in aforementioned journal. However, we aim at adding SEM analysis in all future studies, in all our previous work, we only used the results of TGA for the degradation in addition to XRD and FTIR for characterization. We have added similar suggestions in the manuscript.

doi.org/10.1016/j.jece.2021.105523

Round 2

Reviewer 2 Report

The authors make an effort to address the questions. I agree to publish the manuscript in its current form.